# Unveiling the Texture Secrets of *Morchella* Germplasm: Advanced Grading and Quality Assessment Through Texture Profile Analysis (TPA)

**DOI:** 10.3390/foods14010087

**Published:** 2025-01-01

**Authors:** Jialiang Li, Ying Chen, Xuelian Cao, Jie Tang, Liyuan Xie, Lixu Liu, Yan Wan, Rongping Hu

**Affiliations:** 1Key Laboratory of Coarse Cereal Processing, Ministry of Agriculture and Rural Affairs, Sichuan Engineering & Technology Research Center of Coarse Cereal Industrialization, College of Food and Biological Engineering, Chengdu University, Chengdu 610106, China; 18181788251@163.com; 2Sichuan Institute of Edible Fungi, National Observing and Experimental Station of Agricultural Microbiology, Chengdu 610066, China; garden.daying@163.com (Y.C.); 13547980443@163.com (X.C.); biotang@163.com (J.T.); xieliyuan77@163.com (L.X.); 15198193675@163.com (L.L.); 3Institute of Plant Protection, Sichuan Academy of Agricultural Sciences, Chengdu 610066, China

**Keywords:** morels, texture, grading, quality evaluation

## Abstract

*Morchella* is an edible mushroom with medical applciations. To explore the correlation between the texture indices of *Morchella* and to establish a sensory quality evaluation system, the texture quality characteristics of 214 *Morchella* germplasm resources from our country were analyzed via the Texture Profile Analysis (TPA) method on a texture analyzer. The research revealed significant differences in the texture quality characteristics of both the pileus and stipe among *Morchella* populations. After the Kolmogorov–Smirnov test, the six texture characteristics were seen to conform to a normal distribution. According to the correlation analysis, there was a significant correlation between the texture characteristics of the pileus, and a significant positive correlation between the hardness and the gumminess of the stipe (correlation coefficient of 0.96). Additionally, the hardness was negatively correlated with cohesiveness and resilience, with correlation coefficients of −0.51 and −0.38. Variation analysis indicated abundant genetic variation in each characteristic. Furthermore, the coefficient of variation for the characteristics of *Morchella sextalata* was higher than those of other species. Principal component analysis simplified the texture evaluation indices of *Morchella* pileus into a palatable factor and cohesive factor, and arranged the texture evaluation indexes of *Morchella* stipe into toughness–hardness factor and cohesive factor. Through comprehensive evaluation and cluster analysis, 10 *Morchella* strains were selected for subsequent use as breeding or cultivation materials. By comparing three different methods, the ‘probability grading method’ was determined to be the most suitable evaluation method for the texture quality characteristics of *Morchella*. The research results established a texture evaluation system for *Morchella*, offering a reference for selecting and cultivating breeds with superior texture qualities.

## 1. Introduction

*Morchella esculenta* (L.) Pers., a member of the *Morchella* species within the Pezizales order of Ascomycota [1], is a renowned edible and medicinal mushroom. It is highly favored for its distinctive fragrance, high edibility, medicinal value, and meaty texture. Gourmets also recommend it due to its rich content in terms of essential amino acids and minerals needed by the human body [2,3,4]. Moreover, it contains carbohydrates, proteins, and various bioactive compounds, such as organic acids, polyphenols, and tocopherols [5], which endow it with physiological functions including antioxidant activity [6], immune regulation [7], antitumor properties [8], anti-inflammatory effects [9], and antiviral capabilities [10]. Currently, the large-scale artificial cultivation of *Morchella* is practiced in China, encompassing varieties such as *Morchella importuna*, *Morchella sextelata*, and *Morchella eximia*. The maximum yield of fresh *Morchella* can reach 7.62 metric tons per hectare [11]. However, due to the short harvest window and the diversity of *Morchella* varieties and strains, the quality and texture of fresh *Morchella* are influenced by variations in harvesting, transportation, shelf life, and storage conditions. Thus, it is imperative to conduct an objective assessment of the quality and texture of morel resources and to establish a quality evaluation system, providing a theoretical foundation for the high-value utilization of *Morchella*.

Texture is an important means of measuring the quality of edible products. Traditionally, the texture quality attributes of food are determined via sensory evaluation, a method that is time-consuming, labor-intensive, and costly. Moreover, this approach is heavily dependent on subjective assessments, leading to potential inaccuracies in evaluating food texture quality and providing a scientific basis for researchers. In recent years, the use of texture analyzers in food texture assessment has become increasingly popular, and they are now recognized as standard instruments for testing physical properties [12]. Texture Profile Analysis (TPA), performed using a texture analyzer, simulates the human mouth’s cutting and chewing actions and measures relevant texture parameters, significantly reducing subjective judgment errors in food quality evaluation. This technique has been applied to a variety of foods, including litchi [13], jujube [14], tomato [15,16], peach [17], strawberry [18], and meat [19]. However, no systematic research has been conducted on *Morchella* germplasm resources. Given that fresh *Morchella* has a high water content and a crisp texture, it is prone to damage during storage and distribution. It is therefore essential to study post-harvest texture changes in *Morchella* germplasm resources and to establish a grading index for *Morchella* texture quality. This will aid in selecting breeds with desirable texture characteristics.

To achieve this, this study systematically investigated the variations in texture indices of both the pilei and stipes across different *Morchella* species. It utilized correlation and cluster analyses to explore relationships among *Morchella*’s texture indices, used comprehensive evaluation to screen out morel strains as subsequent breeding or cultivation materials, and identified a grading method appropriate for *Morchella*’s texture indices by evaluating various methods. The Texture Profile Analysis (TPA) method was employed to assess indices including hardness, springiness, cohesiveness, gumminess, chewiness, and resilience. The findings of this research offer a methodological foundation and technical support for evaluating the texture of *Morchella*.

## 2. Materials and Methods

### 2.1. Experiment Materials

A total of 214 cultivated *Morchella* germplasm resources were used in this study (Table 1), including 61 *M. importuna*, 99 *M. sextelata*, and 54 *M. eximia*. These strains were provided by Sichuan Institute of Edible Fungi.

### 2.2. Field Experiment

The field experiment was conducted from 2023 to 2024 at the Modern Agricultural Science and Technology Innovation Demonstration Park of Sichuan Academy of Agricultural Sciences, China (104.12°42.48′ E, 30.46°37.82′ N, 472 m above sea level). *Morchella* was cultivated in a greenhouse, and the sowing method was furrow sowing. The furrow width was 15~20 cm, and the furrow spacing was about 20~30 cm, varying according to the width of the bed surface. *Morchella* was evenly sown in the furrow and covered with 5 cm of soil. The other cultivation management followed the Sichuan *Morchella* high-efficiency cultivation model [20]. Each plot measured 5 m^2^, and there were 3 replicates for each cultivated strain. The changes in temperature, humidity, and carbon dioxide concentration in the greenhouse throughout the experiment are depicted in Figure 1.

According to the ‘Guidelines for the conduct of tests for distinctness, uniformity, and stability—*Morchella*’ (NY/T 4221-2022) [21], 5 fruiting bodies were selected for each cultivated strain, and the indices of the pileus and stipe were measured.

### 2.3. Texture Quality Test Equipment and Test Parameters

The test equipment was a texture analyzer (Model: TA.XT Plus C; Stable Microsystems, Haslemere, Surrey, UK). Given that it is difficult to preserve *Morchella* for extended periods, the samples were harvested and measured on the same day. Using the texture profile analysis (TPA) model, we randomly selected healthy specimens that were growing normally and without disease. The pilei and the stipes of *Morchella* were cut into 1 cm square pieces to evaluate six texture quality indices: hardness, springiness, cohesiveness, gumminess, chewiness, and resilience. The test parameters, determined based on those reported by Lespinard et al. [22]. and Yao et al. [23]. with some improvements, were as follows: a pre-test speed of 1 mm/s, a test speed of 1 mm/s, a post-test speed of 1 mm/s, a target mode strain, a deformation variable of 50%, and a trigger force of 5 g. A P75 (75 mm diameter) column probe was used.

### 2.4. Sensory Evaluation

The sensory evaluation team was composed of members from the Sichuan Institute of Edible Fungi. In accordance with the GB/T 16291.1-2012 ‘*Sensory analysis. General guidance for the selection, training and monitoring of assessors* [24]*. Part 1: Selected assessors*’, a total of 22 sensory evaluators aged between 20 and 40 were selected. These evaluators were chosen based on their strong sensory recognition skills, high sensitivity, and excellent expressive abilities. The team consisted of 12 females and 10 males.

The sensory evaluation indicators for *Morchella* were primarily based on Feng et al. [25]. and Bavay et al. [26], with slight modifications made to better align with the specific characteristics of *Morchella*. To eliminate bias, the *Morchella* samples were randomly coded and boiled in water for 10 min before being evaluated by the reviewers. Evaluators assessed the appearance, taste, and texture of the samples. To avoid sensory fatigue, each sensory evaluation session lasted no longer than 5 min, followed by a 10 min break before the next round. Each tasting session was conducted twice to ensure accuracy and consistency in the evaluation process. The sensory descriptors for *Morchella* were collected, screened, and categorized. After multiple discussions within the sensory evaluation team, the descriptors that best represented the key sensory attributes of *Morchella* were selected. These descriptors were then organized into a sensory scoring table, using a 1–5 point scale, and the judges rated the intensity of each attribute.

### 2.5. Data Analysis

Excel (Microsoft Office 2021) was used to process the original data and calculate the minimum, maximum, range, median, mean (*X*), standard deviation (*S*), and coefficient of variation (*CV*) values of the quality characteristics. SPSS 25.0, GraphPad Prism 10.0, Origin 2021, and chiplot were used to conduct the K-S normality test, significance analysis, box-and-whisker drawing, correlation analysis, cluster analysis, principal component analysis, and a comprehensive evaluation of texture quality.

The coefficient of variation (*CV*) calculation formula is as follows:*CV* (%) = *S*/*X* × 100%(1)

Characteristics were classified using the traditional grading [27] method, the least significant difference (LSD_0.05_) [28], and the probability grading [29] method.

## 3. Results

### 3.1. Significance Analysis of Texture Quality Characteristics

The hardness, springiness, and gumminess values of the pileus of *M. eximia* were significantly higher than those of *M. importuna* and *M. sextelata* (Figure 2D). Among the stipe texture quality characteristics, there were only extremely significant differences in resilience (Figure 2E). The stipe resilience of *M. importuna* was significantly higher than that of other species. The texture quality parameters of the stipe were all higher than those of the pileus. Specifically, the hardness of the stipe was 2–3 times that of the pileus, the springiness and cohesiveness were 1–2 times those of the pileus, the gumminess and resilience were 3–4 times those of the pileus, and the chewiness was 4–5 times that of the pileus.

### 3.2. Analysis of Genetic Variation in Texture Quality Characteristics

The coefficient of variation of the texture quality characteristics of the *Morchella* pileus ranged from 22.4% to 73.9%, with a coefficient of variation greater than 20%, indicating a high degree of variability (Table 2). Among the various populations, the springiness, cohesiveness, gumminess, chewiness, and resilience values of *M. sextelata* were higher than those of other species, reaching 30.8%, 36.7%, 54.3%, 73.9%, and 50.0%, respectively. However, the hardness of *M. importuna* was the highest, with a coefficient of variation of 35.1%. For the other four texture quality characteristics, the coefficients of variation were higher for the pileus than for the stipe, with the average coefficient of variation for stipe hardness being 1.47 times that of the pileus (Table 2 and Table 3). The average coefficients of variation for springiness, cohesiveness, gumminess, chewiness, and resilience of the pileus were 2.33, 2.19, 1.19, 1.44, and 2.84 times those of the stipe, respectively. The averages and medians of springiness, cohesiveness, and resilience for both pileus and stipe texture quality characteristics were the same or similar.

The coefficient of variation of the texture quality characteristics of the *Morchella* stipe ranged from 10% to 53.9% (Table 3), among which hardness, gumminess, and chewiness exhibited the highest degree of variation, with coefficients of variation all exceeding 36%. For *M. sextelata*, the coefficients of variation for hardness, springiness, gumminess, and chewiness were higher than those of other species, at 50.7%, 11.8%, 48.0%, and 45.0%, respectively. Among the species, *M. eximia* had the highest coefficient of variation for cohesiveness, at 14.5%. *M. importuna* had the highest coefficient of variation for resilience, at 18.8%.

### 3.3. Normality Test of Texture Quality Characteristics

The normality test results (Appendix A) indicated that the significance (Sig) values for springiness of the pileus texture quality characteristics of all tested strains were greater than 0.05, suggesting a normal distribution. The significance value for cohesiveness of *M. eximia* (0.200) also suggested a normal distribution. Absolute skewness and kurtosis values less than 1 indicated that the six quality characteristics of *M. eximia*’s pileus, the five characteristics of *M. sextelata*’s pileus (excluding chewiness), and the springiness, cohesiveness, and resilience of *M. importuna*’s pileus displayed a normal distribution. The box-and-whisker plot (Figure 3) showed that, excluding the outliers, the hardness, gumminess, and chewiness values of *M. importuna*’s pileus, as well as the chewiness of *M. sextelata*’s pileus, were approximately normally distributed.

As shown in Figure 3, no outliers were observed in the hardness of the stipe, the chewiness of *M. importuna*, the gumminess and chewiness of *M. sextelata*, and the resilience of *M. eximia* [30], suggesting that the data can be considered normally distributed. Appendix A indicates that the texture quality characteristics of the stipes with Sig values greater than 0.05 include the springiness, cohesiveness, and resilience of both *M. importuna* and *M. sextelata*, indicating a normal distribution. Absolute skewness and kurtosis values less than 1 were observed for the cohesiveness, gumminess, and chewiness of *M. eximia*, which were approximately normally distributed.

### 3.4. Correlation Analysis and Cluster Analysis of Texture Quality Characteristics

Correlation analysis showed significant positive correlations (*p* < 0.01) between the hardness, springiness, cohesiveness, gumminess, chewiness, and resilience of the *Morchella* pileus, with correlation coefficients all above 0.19 (Figure 4A). Significant correlation (*p* < 0.01) was observed between the hardness, cohesiveness, and gumminess of the stipe (Figure 4B). Specifically, there was a significant positive correlation between springiness and gumminess, and a significant negative correlation between springiness and resilience. Additionally, a significant positive correlation was found between springiness and both cohesiveness and chewiness, as well as between gumminess and resilience. The respective correlation coefficients were 0.61, 0.13, 0.36, and 0.37. The correlation coefficient between chewiness and resilience was −0.13, indicating a significant negative correlation.

The cluster heatmaps shown in Figure 4C,D indicate that the six texture indices of both the pileus and stipe of *Morchella* can be categorized into three distinct groups. Group A of the *Morchella* pileus was used to assess hardness, gumminess, and chewiness. Group B assessed resilience, and Group C considered springiness and cohesiveness. According to the classification, Group A shows the hardness factor, Group B shows the resilience factor, and Group C shows the toughness factor. The classification indicated that the factors defined by the stipe and pileus of *Morchella* were consistent.

The 214 tested strains’ pilei and stalks were analyzed using cluster heatmaps (Figure 4C,D). According to the clustering results, each could be categorized into three groups. The cluster analysis revealed that the six texture indices in Group A were higher than those in the other groups, with *M. importuna* and *M. eximia* comprising about 35% of this group. In Group B, these indices were lower, with *M. sextelata* accounting for 72.1% of the group. Group C was characterized by high springiness and cohesiveness but low hardness, gumminess, and chewiness. Given that gumminess equals hardness times cohesiveness, and chewiness equals springiness times gumminess, strains with high springiness and cohesiveness but low hardness may exhibit low gumminess and chewiness, as observed in Group C. For the stalk, the cluster analysis showed that the hardness, gumminess, and chewiness in Group D were higher than in the other groups, with *M. sextelata* and *M. eximia* each making up about 40% of this group. Group E had the lowest springiness index, with *M. sextelata* showing a value of 51.9%. Group F exhibited the highest means for cohesiveness and resilience, with *M. sextelata* also having the highest proportion, at 41.0%.

Based on the normal distribution results shown in Figure 3A–F, M15, M154, and M209 were identified as representative strains of Group A (Figure 2), with their mean values of gumminess and chewiness indices being among the highest, as shown in Figure 3. M17, M101, and M210 were identified as representative strains of Group B, with the means of one or more texture indices being among the lowest in Figure 3. M33, M100, and M186, representative strains of Group C, had high mean springiness and cohesiveness indices in Figure 3, while their mean hardness index was low. Representative strains of Group D, M55, M99, and M166 (Figure 2), were identified, and their average gumminess index, shown in Figure 3, was the highest. Representative strains of Group E, M56, M141, and M178, were identified, with their average springiness index consistently low. This is shown in Figure 3. Representative strains of Group F, M32, M80, and M180, were identified, and their average indices of cohesiveness and resilience in Figure 3 were both high.

### 3.5. Correlations Between Instrumental Measurements and Sensory Evaluation of Morchella

To verify the correlation between the instrument texture indices and consumer perception, it is important to note that the stipe of *Morchella* is typically used as a processed product, while the pileus is used as a fresh product. Therefore, sensory evaluation data for the pileus of 18 representative *Morchella* strains (Figure 2) were analyzed in conjunction with texture parameters, measured by a texture analyzer, using Pearson correlation analysis. This analysis aimed to explore the relationship between the instrument-measured parameters and the sensory analysis results, with the goal of accurately and intuitively reflecting the sensory quality of morel through instrumental testing. As shown in Table 4, there is a correlation between the instrumental texture indices of *Morchella* and the sensory evaluation data. Specifically, the texture hardness index is extremely significantly correlated with sensory crumbliness and hardness (*p* < 0.01) and is significantly correlated with chewiness (*p* < 0.05). The texture springiness index shows a very significant correlation with sensory springiness (*p* < 0.01) and a significant correlation with sensory chewiness (*p* < 0.05). The texture cohesiveness index is significantly correlated with both sensory springiness and chewiness (*p* < 0.05). The texture gumminess index and chewiness index are extremely significantly correlated with sensory crumbliness and springiness (*p* < 0.01). However, the texture resilience index shows no significant correlation with any of the sensory evaluation data. The typical correlation coefficient was further extracted, with the highest value being 0.997 (Table 5). This reached a very significant level (*p* < 0.01). This indicates the existence of a very strong correlation between the instrumental texture indices and sensory data. Therefore, it is possible to use the instrumental texture indices to effectively reflect the sensory qualities of *Morchella*.

### 3.6. Principal Component Analysis and Comprehensive Evaluation of Texture Quality Characteristics

Principal component analysis was performed on the six texture indices of the pileus and stipe of *Morchella* (Table 6 and Table 7). With a cumulative contribution rate greater than 80% or an eigenvalue greater than 1 as the standard, two principal components were selected to represent most of the information of the six texture indices. The contribution rate of the first principal component of the pileus was 71.5%, and the representative indicators were gumminess and chewiness. These indicators reflected the palatability of *Morchella* and were defined as palatable factors. The contribution rate of the second principal component of the pileus was 14.0%, and the representative indicator was cohesiveness, which reflected the size of the aggregation force between the cells of the pileus of *Morchella* and could be defined as the cohesive factor; the contribution rate of the first principal component of the stipe was 59.4%, and the representative indicators were hardness and gumminess, which could reflect the quality characteristics of *Morchella* and were defined as the toughness–hardness factor. The contribution rate of the second principal component of the stipe was 30.1%, and the representative indicators were springiness and cohesiveness, which were defined as the cohesive factors. Through principal component analysis, the texture evaluation indices of *Morchella*’s pileus were finally simplified into a palatable factor and cohesive factor, and the texture evaluation indices of *Morchella*’s stipe were simplified into a toughness–hardness factor and a cohesiveness factor.

According to the principal component loading coefficients of each texture index and the corresponding eigenvalues, two principal component expressions are constructed: *F*1 = 0.363x1 + 0.365x2 + 0.415x3 + 0.469x4 + 0.471x5 + 0.349x6, *F*2 = −0.685x1 + 0.131x2 + 0.512x3 − 0.205x4 − 0.142x5 + 0.434x6, *F*3 = 0.514y1 − 0.077y2 − 0.391y3 + 0.488y4 + 0.463y5 − 0.351y6, *F*4 = 0.147y1 + 0.653y2 + 0.355y3 + 0.258y4 + 0.355y5 + 0.400y6. Here, x and y represent the values of the pileus and stipe texture indices in the standardized range of (0, 1). The ratio of the variance contribution rate of each principal component to the cumulative variance contribution rate is used as the weight needed to establish a comprehensive evaluation model: *F*c = 0.715F1 + 0.140F2, *F*s = 0.594F3 + 0.301F4. The higher the F value, the better the overall performance [31]. As shown in Appendix A, the comprehensive score of *Morchella*’s pileus ranges from 0.044 to 1.630, and the top five strains are M209, M192, M15, M14, and M172. The texture characteristics of the pileus of this type of strain are high cohesiveness, and they are all in Group A in the cluster analysis; according to Appendix A, the comprehensive score of *Morchella*’s stipe ranges from −0.087 to 1.079, and the top five strains with the highest scores are M147, M161, M99, M67, and M166. The texture characteristics of the stipe of this type of strain are high hardness, stickiness, and chewiness, and they are all in Group D in the cluster analysis.

### 3.7. Classification of Texture Quality Characteristics

After comparing the grading methods for the texture quality characteristics of *Morchella* pileus and stipes (Appendix A), it was found that some characteristics in the traditional grading method could only be classified as belonging to the highest level. In the least significant difference method, some characteristics had no distribution among the strains. The probability grading method, which provided data on both properties and grading ranges, was divided into either 3 or 5 levels.

## 4. Discussion

This study analyzed six indices of the pileus and stipes of different *Morchella* species via TPA testing using a texture analyzer. These indices included hardness, which simulates the squeezing force exerted by human teeth on the sample; springiness, simulating the height recoverable from the first to the beginning of the second tooth compression; cohesiveness, referring to the binding force between tissue cells; gumminess, testing the sample’s viscosity; chewiness, reflecting the sample’s resistance to chewing; and resilience, referring to the sample’s ability to recover its shape after compression. The stipe of *Morchella*, considered an industrial byproduct [32], showed significantly higher TPA-evaluated texture indices than the pileus, suggesting that it might offer a richer taste, thus providing a theoretical basis for its use in food processing, such as in canned food, condiments, and vegetarian meat products. Correlation analysis revealed that the hardness of *Morchella* was positively correlated with its springiness and chewiness. This suggests that the higher the hardness of the *Morchella*, the denser its flesh is likely to be, making it more springy and more resistant to both chewing and storage. These findings align with existing research on fruits and other foods [33,34]. Since *Morchella* is fragile and prone to damage during transportation, the higher its hardness, the better it can maintain its overall quality and integrity throughout the process. The texture analyzer test revealed that *M. eximia* had the greatest hardness, springiness, and chewiness, indicating that it may be more suitable for storage and transportation than the pileus of *M. importuna* and *M. sextelata*.

The coefficient of variation reflects the differences in the evolutionary conservatism and genetic plasticity of quantitative traits, and a higher coefficient of variation is more conducive to the classification of indices [35,36]. The surface of the *Morchella* pileus had characteristics such as pits and ribbed structures, while the surface of the stipe was relatively smooth. Therefore, the coefficient of variation of the six texture indices of the pileus in this study was larger and wider than that of the stipe, ranging from 22.4% to 73.9%, indicating that the degree of variation in pileus characteristics was richer, which was conducive to the genetic improvement of germplasm resources. From the results of the comparison of coefficients of variation among *Morchella* species, the coefficient of variation of *Morchella sextelata* was larger than that of *Morchella eximia* and *Morchella importuna*, with the range of the pileus being 30.8% to 73.9% and the range of the stipe being 11.8% to 53.9%, indicating that the variation in *M. sextelata* was more substantial. However, this phenomenon might also be due to the difference caused by the larger sample size of *M. sextelata*. Correlation analysis found that the six texture indices of the *Morchella* pileus showed an extremely significant positive correlation, which might be due to the hollow of the *Morchella* cap and the unique uneven surface characteristics of the pileus. In the *Morchella* stipe, hardness had an extremely significant negative correlation with cohesiveness, and an extremely significant positive correlation with gumminess, which was consistent with the trend of the research results of Kong et al. [14] for winter jujube. The results of the correlation analysis further showed that there was an overlap of information between different texture indices [37]. Therefore, the texture evaluation of *Morchella* can be effectively enhanced by screening texture evaluation indices.

According to the cluster analysis of the 214 tested strains, it was found that in the classification groups with low texture parameters, specifically group b and group e, the proportion of *Morchella sextelata* was significantly higher than that of the other two species. Conversely, in the classification groups with the highest hardness parameters, namely, Group A and Group D, the proportion of *Morchella eximia* was higher, which was consistent with the average value data reported in the previous article. Furthermore, cluster analysis of the tested strains is beneficial for understanding the texture characteristics of different varieties of *Morchella* based on their variety group. This not only provides a variety reference for the cultivation of *Morchella* in production but also offers a basis for the selection in genetic resource breeding. A typical correlation analysis was conducted between the sensory evaluation data of *Morchella* and the texture parameters measured by the texture analyzer. The results of the correlation analysis revealed a very significant correlation between the texture parameters and sensory data. Therefore, texture parameters can effectively reflect the textural sensory qualities of *Morchella*.

To make a perform comprehensive and accurate evaluation of food quality, people often use a variety of texture indices. However, when there are too many indices, the information between them can overlap, thus affecting the accuracy of the analysis results. Since gumminess is represented by the product of hardness and cohesiveness, and as chewiness is represented by the product of springiness and gumminess, when hardness, chewiness, and gumminess are considered together, hardness is selected as the representative index. Cohesiveness is the cohesive force within *Morchella*, defined as the ratio of the positive force of the probe in the second compression cycle to that in the first cycle in the TPA (Texture Profile Analysis) test. Therefore, the texture evaluation indices of *Morchella* were finally simplified into four categories according to the results of principal component analysis: we considered palatability factor and cohesiveness factor for the pileus, and toughness–hardness factor and cohesiveness factor for the stipe. Based on the ranking of the comprehensive evaluation *F* value, the top five strains of *Morchella*’s pileus scores were M209, M192, M15, M14, and M172, and the pilei of these strains had the texture characteristics of high cohesiveness and springiness. The top five strains of *Morchella*’s stipe scores were M147, M161, M99, M67 and M166, and the stipes of these strains had the texture characteristics of high hardness, gumminess, and chewiness, but low cohesiveness. This was consistent with the results of cluster analysis and the stipes were in Group A or Group D, respectively. The selected strains can be further utilized as breeding materials or high-quality germplasm resources.

The probability classification method can objectively display the variation characteristics of quantitative characteristics, thus providing a more unified standard for their classification [14]. Since texture quality characteristics generally conform to the normal distribution and as the median and mean values of most traits are similar, the probability classification method was chosen as the classification method for the texture quality characteristics of *Morchella* after a comparison. All six texture indices can be successfully classified using the probability classification method. This method has been applied to the quantitative characteristics of various crops such as *Pleurotus giganteus* [38], *Canarium album* [39], and *Chrysanthemum* [40]. In this study, 214 samples of three typical cultivated *Morchella* species were classified, which fully represented the texture characteristics of *Morchella* species cultivated in China.

## 5. Conclusions

The pileus and stipe of three cultivated species of *Morchella* were tested using TPA (Texture Profile Analysis). It was found that the texture parameters of the stipe of *Morchella* were higher than those of the pileus, suggesting that the stipe may offer a richer taste. This provides a basis for its potential application in the field of food processing. Through comparison and correlation analysis, it was observed that the hardness of *M. eximia* was positively correlated with its springiness and chewiness, with these indices also being the highest. These findings can serve as a reference for selecting species to enhance the transportation and storage duration of fresh *Morchella*. Canonical correlation analysis between the instrumental texture indices of *Morchella* and the sensory evaluation data confirms that the instrumental texture parameters can effectively reflect consumers’ perceptions of fresh *Morchella*. The coefficients of variation of the texture indices for both the pileus and stipe of *Morchella* were large, and the correlations were strong. Then, principal component analysis was performed on each index, the evaluation index of *Morchella*’s pileus texture was simplified into a palatable factor and cohesive factor, and the evaluation index of *Morchella*’s stipe texture was simplified into a toughness–hardness factor and cohesive factor. Based on cluster analysis and comprehensive evaluation, 10 *Morchella* strains were selected for further use as breeding or cultivation materials. Finally, by comparing different classification methods, the probability classification method was selected to provide a standard for the texture parameters of cultivated *Morchella* in China.

## Figures and Tables

**Figure 1 foods-14-00087-f001:**
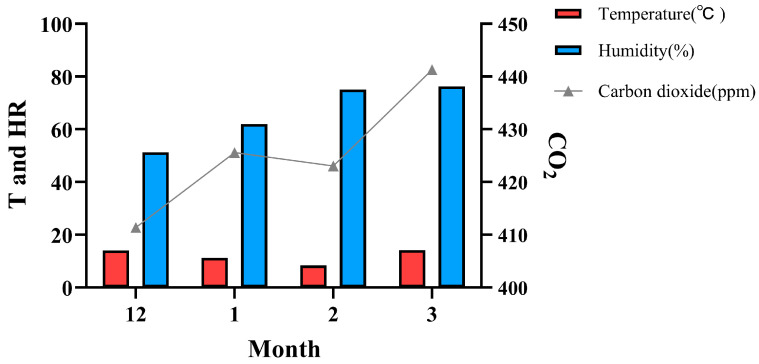
Climate change diagram in greenhouse.

**Figure 2 foods-14-00087-f002:**
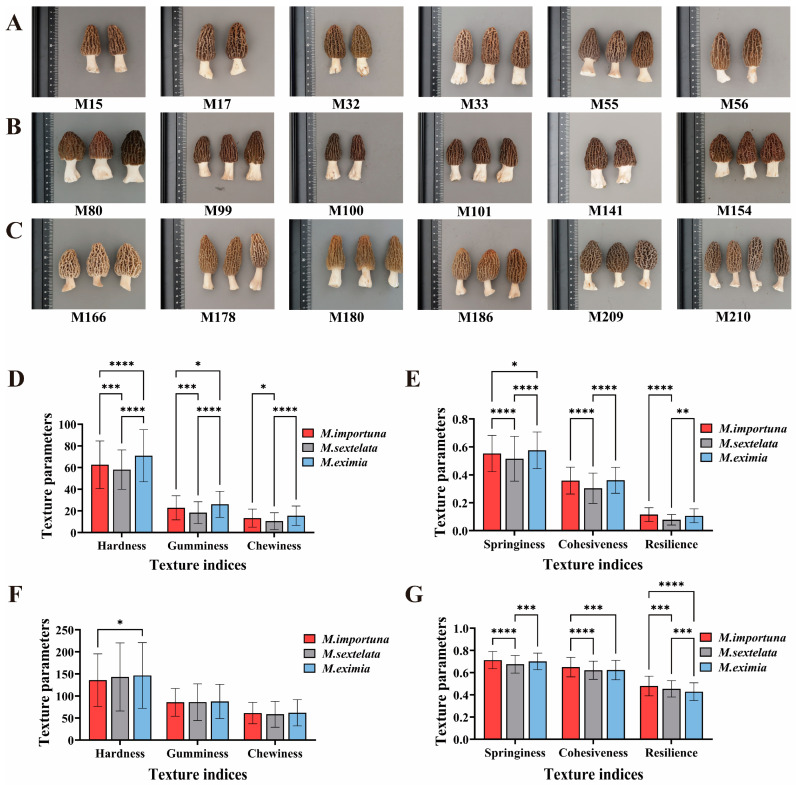
Representative strains and significance analysis of texture parameters of *Morchella*: (**A**) represents the representative strains of *M. importuna* from different groups in strain clustering analysis; (**B**) represents the representative strains of *M. sextelata* from different groups in strain clustering analysis; (**C**) represents the representative strains of *M. eximia* from different groups in strain clustering analysis; (**D**,**E**) are the texture parameters of the pileus of different species of *Morchella*; (**F**,**G**) are the texture parameters of the stipes of different species of *Morchella*. **** indicates extremely significant correlation (*p* < 0.0001), *** indicates extremely significant correlation (*p* < 0.001), ** indicates extremely significant correlation (*p* < 0.01), and * indicates significant correlation (*p* < 0.05).

**Figure 3 foods-14-00087-f003:**
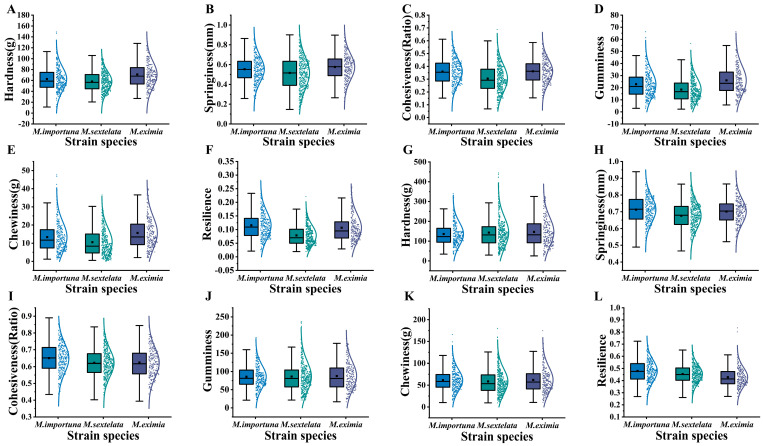
Box-and-whisker analysis of texture parameters of *Morchella*. (**A**–**F**) are the texture indices of the *Morchella* pileus. (**G**–**L**) are the texture indices of the *Morchella* stipe. The colored boxes in each figure represent different types of *Morchella*, while each point indicates the value of a sample.

**Figure 4 foods-14-00087-f004:**
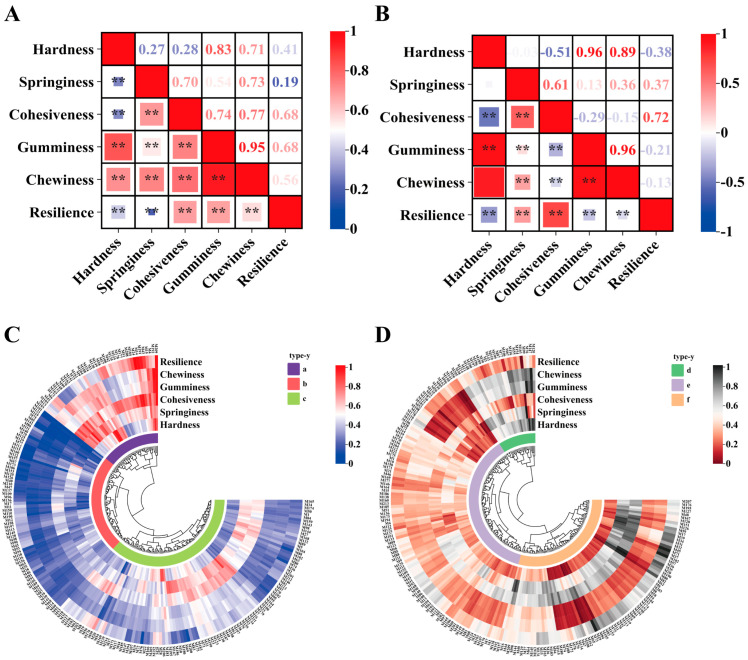
Significance and clustering analysis of texture parameters of *Morchell*: (**A**,**B**), respectively, represent the correlation between the pileus and stipe texture indices of all species of *Morchella*. ** indicates extremely significant correlation (*p* < 0.01) (**C**,**D**) are cluster heatmaps of the pileus and stipe of all species of *Morchella*. The clustering method was applied to group the six texture indicators of 214 strains. The pileus and stipes of *Morchella* were categorized into three groups for further analysis: these were groups a, b, and c for the pileus, and groups d, e, and f for the stipes.

**Table 1 foods-14-00087-t001:** Test strains.

NO.	Strains	Species	Source
1	M1–M61	*M. importuna*	Chengdu City, Sichuan Province
2	M62–M72,M86–M106,M110–M111,M115–M119,M125,M128–M160	*M. sextelata*	Chengdu City, Sichuan Province
3	M73–M74		Shiyan City, Hubei Province
4	M75,M112		Mianyang City, Sichuan Province
5	M76		Zunyi City, Guizhou Province
6	M77		Dazhou City, Sichuan Province
7	M78		Hanzhong City, Shanxi Province
8	M79		Taiyuan City, Shanxi Province
9	M80–M81		Lanzhou City, Gansu Province
10	M82–M83,M114		Nanyang City, Henan Province
11	M84–M85		Neijiang City, Sichuan Province
12	M107–M109		Jinan City, Shandong Province
13	M113		Guangzhou City, Guangdong Province
14	M120		Changsha City, Hunan Province
15	M121–M123		Suzhou City, Anhui Province
16	M124		Kunming City, Yunnan Province
17	M126		Bazhong City, Sichuan Province
18	M127		Shenyang City, Liaoning Province
19	M161,M166–M175,M177–M178,M180,M182–M185,M187–M197,M199–M214	*M. eximia*	Chengdu City, Sichuan Province
20	M162,M165		Hanzhong City, Shanxi Province
21	M163		Yibin City, Sichuan Province
22	M164		Kangding City, Sichuan Province
23	M176		Urumqi City, Xinjiang Uygur Autonomous Region
24	M179		Mianyang City, Sichuan Province
25	M181		Shenyang City, Liaoning Province
26	M186		Kunming City, Yunnan Province
27	M198		Bazhong City, Sichuan Province

**Table 2 foods-14-00087-t002:** Variation analysis of six characteristics of *Morchella* pileus.

Characteristic	Strain Species	Min	Max	Range	Median	Mean	Standard Deviation	Variation Coefficient/%
Hardness	*M. importuna*	11.12	149.06	137.94	58.71	62.61	21.97	35.09
	*M. sextelata*	20.50	113.86	93.36	56.48	58.12	18.19	31.30
	*M. eximia*	26.82	147.61	120.79	67.8	70.99	24.06	33.89
Springiness	*M. importuna*	0.26	0.94	0.68	0.55	0.55	0.13	23.64
	*M. sextelata*	0.15	0.90	0.75	0.52	0.52	0.16	30.77
	*M. eximia*	0.23	0.92	0.69	0.58	0.58	0.13	22.41
Cohesiveness	*M. importuna*	0.15	0.61	0.46	0.35	0.36	0.10	27.78
	*M. sextelata*	0.07	0.69	0.62	0.29	0.30	0.11	36.67
	*M. eximia*	0.15	0.62	0.47	0.36	0.36	0.09	25.00
Gumminess	*M. importuna*	2.94	66.28	63.34	21.02	22.89	11.14	48.67
	*M. sextelata*	2.28	56.56	54.28	16.68	18.42	10.00	54.29
	*M. eximia*	5.71	66.36	60.65	23.61	26.08	12.03	46.13
Chewiness	*M. importuna*	1.19	47.61	46.42	11.69	13.35	8.34	62.47
	*M. sextelata*	0.52	36.54	36.02	8.33	10.53	7.78	73.88
	*M. eximia*	2.05	44.18	42.13	13.42	15.64	8.93	57.10
Resilience	*M. importuna*	0.02	0.28	0.26	0.11	0.12	0.05	41.67
	*M. sextelata*	0.02	0.22	0.20	0.07	0.08	0.04	50.00
	*M. eximia*	0.03	0.26	0.23	0.10	0.11	0.05	45.45

**Table 3 foods-14-00087-t003:** Variation analysis of six characteristics of *Morchella* stipe.

Characteristic	Strain Species	Min	Max	Range	Median	Mean	Standard Deviation	Variation Coefficient/%
Hardness	*M. importuna*	34.27	340.18	305.91	122.98	135.95	59.53	43.79
	*M. sextelata*	29.32	444.19	414.87	131.16	143.16	77.11	53.86
	*M. eximia*	25.25	348.22	322.97	131.95	146.55	74.34	50.73
Springiness	*M. importuna*	0.48	0.94	0.46	0.72	0.71	0.08	11.27
	*M. sextelata*	0.42	0.87	0.45	0.68	0.68	0.08	11.76
	*M. eximia*	0.50	0.91	0.41	0.70	0.70	0.07	10.00
Cohesiveness	*M. importuna*	0.40	0.89	0.49	0.65	0.65	0.09	13.85
	*M. sextelata*	0.39	0.86	0.47	0.62	0.62	0.08	12.90
	*M. eximia*	0.39	0.84	0.45	0.62	0.62	0.09	14.52
Gumminess	*M. importuna*	21.20	193.03	171.83	81.28	85.70	31.58	36.85
	*M. sextelata*	21.45	236.45	215.00	80.90	86.08	41.34	48.03
	*M. eximia*	16.69	194.29	177.60	80.79	87.70	38.60	44.01
Chewiness	*M. importuna*	10.09	165.90	155.81	58.59	61.42	24.33	39.61
	*M. sextelata*	9.34	179.56	170.22	53.45	58.54	29.25	49.97
	*M. eximia*	10.45	174.56	164.11	57.11	62.04	29.76	47.97
Resilience	*M. importuna*	0.27	0.73	0.46	0.48	0.48	0.09	18.75
	*M. sextelata*	0.24	0.68	0.44	0.45	0.45	0.07	15.56
	*M. eximia*	0.27	0.83	0.56	0.41	0.43	0.08	18.60

**Table 4 foods-14-00087-t004:** Correlation between sensory evaluation indices and texture parameters.

	Sensory	Umami	Acceptability	Crumbliness	Hardness	Springiness	Chewiness
Instrument							
Hardness		0.099	0.037	−0.716 **	0.842 **	0.352	0.569 *
Springiness		0.158	0.146	−0.331	0.326	0.882 **	0.551 *
Cohesiveness		−0.011	0.188	−0.358	0.455	0.560 *	0.516 *
Gumminess		0.120	0.033	−0.619 **	0.723 **	0.425	0.581 *
Chewiness		0.179	0.033	−0.650 **	0.661 **	0.558 *	0.639 **
Resillience		−0.194	0.254	−0.439	0.418	0.269	0.324

Note: ** indicates extremely significant correlation (*p* < 0.01); * indicates significant correlation (*p* < 0.05).

**Table 5 foods-14-00087-t005:** Canonical correlation coefficients between texture parameters and sensory indices of *Morchella*.

No.	Canonical Correlation Coefficient	Significance
1	0.997	0.000
2	0.888	0.069
3	0.826	0.180
4	0.746	0.410
5	0.276	0.931
6	0.044	0.886

**Table 6 foods-14-00087-t006:** Principal component analysis of texture parameters of *Morchella* pileus.

Characteristic	Feature Vector	
	First Principal Component	Second Principal Component
Hardness	0.752	−0.628
Springiness	0.755	0.120
Cohesiveness	0.860	0.469
Gumminess	0.971	−0.188
Chewiness	0.975	−0.130
Resilience	0.723	0.398
Characteristic value	4.289	0.840
Contribution rate	71.481	13.993
Cumulative contribution rate/%	71.481	85.474

**Table 7 foods-14-00087-t007:** Principal component analysis of texture parameters of *Morchella* stipe.

Characteristic	Feature Vector	
	First Principal Component	Second Principal Component
Hardness	0.970	0.197
Springiness	−0.145	0.877
Cohesiveness	−0.738	0.600
Gumminess	0.922	0.346
Chewiness	0.874	0.477
Resilience	−0.663	0.537
Characteristic value	3.563	1.803
Contribution rate	59.379	30.051
Cumulative contribution rate/%	59.379	89.430

## Data Availability

The original contributions presented in this study are included in the article/Appendix A. Further inquiries can be directed to the corresponding authors.

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
