# Peer review of "Unveiling the Texture Secrets of Morchella Germplasm: Advanced Grading and Quality Assessment Through Texture Profile Analysis (TPA)"

_foods, 2025, doi:10.3390/foods14010087_

Round 1
Reviewer 1 Report
Comments and Suggestions for Authors
The manuscript presents a well-executed study on the texture evaluation of Morchella germplasm resources using Texture Profile Analysis (TPA) and statistical methods. However, there are areas where the manuscript could be improved to enhance clarity, accessibility, and overall impact.
1. Provide more details on sample preparation, as this may influence the TPA results: How were the pileus and stipe samples selected? Were the samples standardized for size, weight, and moisture content? What were the storage conditions prior to testing?
2. Inclusion of sensory evaluation would validate the relevance of the measured texture indices to consumer perception.
3. Analysis of post-harvest changes (e.g., during storage under controlled conditions) should be considered.
4. Some tables, such as Tables 4 and 5 (normality tests), provide excessive detail that may overwhelm readers; move detailed results to supplementary materials and summarize key findings in the main text.
5. Figure 3 and 4 are informative but dense. Add clear annotations or brief explanations to guide readers in interpreting these figures.
Reviewer 2 Report
Comments and Suggestions for Authors
Dear Authors, you should address my comments highlighted across the text.
